# In Vitro Effect of Estrogen and Progesterone on Cytogenetic Profile of Uterine Leiomyomas

**DOI:** 10.3390/ijms26010096

**Published:** 2024-12-26

**Authors:** Alla S. Koltsova, Anna A. Pendina, Olga V. Malysheva, Ekaterina D. Trusova, Dmitrii A. Staroverov, Maria I. Yarmolinskaya, Nikolai I. Polenov, Andrey S. Glotov, Igor Yu. Kogan, Olga A. Efimova

**Affiliations:** D.O. Ott Research Institute of Obstetrics, Gynecology and Reproductology, 199034 St. Petersburg, Russia; rosenrot15@yandex.ru (A.S.K.);

**Keywords:** uterine leiomyoma, abnormal karyotype, intratumoral heterogeneity, clonal composition, tumor evolution, complex chromosomal rearrangements, genomic index, estrogen, progesterone, cell culture

## Abstract

In the present study, we aimed to investigate intratumoral karyotype diversity as well as the estrogen/progesterone effect on the cytogenetic profile of uterine leiomyomas (ULs). A total of 15 UL samples obtained from 15 patients were cultured in the media supplemented with estrogen and/or progesterone and without adding hormones. Conventional cytogenetic analysis of culture samples revealed clonal chromosomal abnormalities in 11 out of 15 ULs. Cytogenetic findings were presented by simple and complex chromosomal rearrangements (64% and 36% of cases, respectively) verified through FISH and aCGH. In most ULs with complex chromosomal rearrangements, the breakpoints did not feature clusterization on a single chromosome but were evenly distributed across rearranged chromosomes. The number of breakpoints showed a strong positive correlation with the number of rearranged chromosomes. Moreover, both abovementioned parameters were in a linear dependency from the number of karyotypically different clones per UL. This suggests that complex chromosomal rearrangements in ULs predominantly originate through sequential events rather than one hit. The results of UL cytogenetic analysis depended on the presence of estrogen and/or progesterone in the culture medium. The greatest variety of cytogenetically different cell clones was detected in the samples cultured without hormone supplementation. Their counterparts cultured with progesterone supplementation showed a sharp decrease in clone number, whereas such a decrease induced by estrogen or estrogen–progesterone supplementation was insignificant. These findings suggest that estrogen–progesterone balance is crucial for forming a UL cytogenetic profile, which, in turn, may underlie the unique response of the every karyotypically abnormal UL to medications.

## 1. Introduction

Uterine leiomyomas (ULs), also known as uterine fibroids, are benign hormone-dependent tumors of myometrial origin. The estimated UL incidence in childbearing age women is 70–80%, with ~40% of patients reporting a severe impact on their quality of life. Considering the fairly poor efficiency of medications, UL often becomes an indication for surgery. Therefore, fundamental mechanisms of UL pathogenesis merit extensive study to define promising treatment targets.

The definitive cause of ULs is unknown but commonly assumed to be associated with hereditary predisposition, lifestyle, and adverse environmental exposure throughout different ontogenetic stages [1,2,3,4,5,6,7]. Spontaneous genetic and epigenetic alterations in myometrial cells contribute to disease development [8,9,10,11,12]. Investigators have reported numerous UL-specific genetic disorders affecting genes and chromosomes. In total, over 200 chromosomal abnormalities have been identified in ULs, mostly represented by simple rearrangements with two breakpoint junctions involving chromosomal regions 1p, 1q, 3q, 6p21, 7q, 10q22, 12q15, and 14q24 [13]. UL chromosomal abnormalities show a significant incidence of complex chromosomal rearrangements, including chromothripsis as an extreme genomic event detected in ULs [14,15,16,17,18]. Considering that different chromosomal abnormalities in ULs present specific pathomorphological and clinical patterns [19,20,21], as well as variable responses to hormone therapy [22,23,24], the contribution of karyotypic anomalies to UL pathogenesis seems justified.

The interpretation of cytogenetic results in ULs is challenged by intratumoral heterogeneity [1,25]. Alongside chromosomally abnormal cells, ULs also contain 46,XX cells, suggesting the secondary origin of karyotypic changes during tumor growth [15,17,26,27]. However, our overall knowledge of the emergence of chromosomal abnormalities and their impact on UL cell sensitivity to external stimuli, especially hormones, remains critically scarce.

The study objective was to investigate intratumoral karyotype diversity, as well as the estrogen/progesterone impact on the UL cytogenetic profile. Conventional karyotyping combined with metaphase fluorescence in situ hybridization (FISH) allowed us to characterize chromosomal abnormalities and the clonal composition of UL samples cultured without hormone supplementation and in the media supplemented with estrogen, progesterone, or both hormones. Our results shed light on the evolutionary patterns associated with the observed UL karyotypic heterogeneity, the processes of structural chromosomal rearrangement formation in ULs, and whether karyotype differences among the cells obtained from the same UL can affect their sex steroid hormone sensitivity.

## 2. Results

### 2.1. Cytogenetic Characteristics of UL Cultures Depend on the Presence of Estrogen and/or Progesterone in the Culture Medium

To assess the effect of sex steroid hormones on the spectrum of chromosomal abnormalities in UL cultures, we investigated the metaphase chromosomes from UL cells cultured with and without hormone supplementation. Four cell cultures were obtained from each of the fifteen ULs: (i) in standard conditions without hormone supplementation, (ii) with estrogen supplementation, (iii) with progesterone supplementation, and (iv) with estrogen and progesterone supplementation. Thus, a total of 60 cell culture samples were yielded from 15 ULs.

Conventional cytogenetic analysis of the 15 UL samples cultured in standard conditions without sex hormone supplementation identified 4 samples with a normal female karyotype (46,XX) for all analyzed cells (UL1–4), whereas the remaining 11 culture samples showed chromosomal abnormalities, such as translocations in 3 cases (UL6–8), deletions of the long arm of chromosome 7 (7q) in 4 cases (UL9–12), and complex chromosomal rearrangements involving three or more breakpoints in the other 4 cases (UL5, 13–15). FISH was performed to specify the structure of rearranged chromosomes. Of 11 UL culture samples with chromosomal abnormalities, 5 demonstrated co-existence of one karyotypically abnormal cell clone with karyotypically normal cells (UL5–8, 12). Another four UL culture samples (UL9–11, 14) revealed three cell clones each, one with normal and two with abnormal karyotype showing different chromosomal abnormalities. One UL culture sample (UL13) revealed two karyotypically abnormal cell clones. Finally, in one UL culture sample (UL15), four karyotypically abnormal cell clones were observed co-existing with 46,XX cells. Thus, in most UL samples cultured without sex steroid hormone supplementation, conventional karyotyping detected clones with abnormal karyotypes predominantly composed of structural chromosomal alterations.

For each of the 15 ULs, conventional cytogenetic analysis of samples cultured with hormone supplementation was also performed, and their karyotype was compared with the corresponding samples cultured without hormones (Table 1). In 6 (UL1–6) of the 15 ULs (40%), no cytogenetic differences were observed between culture samples. The latter included four cases (UL1–4) with a normal karyotype in all culture samples, and two cases (UL5, 6) presented karyotypically normal and karyotypically abnormal cells in all culture samples. The other nine ULs (UL7–15) (60%) showed cytogenetically discordant clonal composition between UL samples cultured with and without hormone supplementation (Table 1). Each of the nine ULs presented with at least one karyotypically abnormal cell clone in the sample cultured without hormone supplementation, whereas cytogenetic findings for samples cultured with estrogen, progesterone, or both hormones showed elimination of at least one clone in either of the three hormone-supplemented cultures. In two of these ULs (UL10, 15), samples cultured with hormone supplementation also showed chromosomal abnormalities that were not detected in standard culture conditions. Notably, none of the UL samples cultured without hormone supplementation demonstrated a normal karyotype in all analyzed cells, whereas corresponding UL samples cultured with hormone supplementation presented an abnormal karyotype. One case (UL8), however, showed the opposite pattern, as the sample cultured without hormones presented cells with normal as well as abnormal karyotypes, while the samples cultured with hormone supplementation were characterized by a normal karyotype in all analyzed cells (Table 1). Thus, in 60% of ULs, conventional cytogenetic analysis showed discordant clonal composition between samples cultured with and without sex steroid hormone supplementation.

To evaluate the effect of estrogen, progesterone, or the combination of both hormones on the clonal composition of UL cultures, the total number of karyotypically different cell clones was compared between UL culture samples grown in different media. In a total of 60 culture samples derived from 15 ULs, 33 karyotypically different clones were detected through conventional cytogenetic analysis (Table 1). The utmost number of clones (30, i.e., 91% of all identified clones) was observed in samples cultured with no hormone supplementation. Samples cultured with hormone supplementation showed 23 clones for samples from the “estrogenic” medium, 21 clones for samples from progesterone-containing medium, and 26 clones for samples cultured with the supplemented estrogen–progesterone combination. Fischer’s exact test showed significant decrease in clone number among samples cultured with progesterone supplementation, compared to their counterparts cultured with no hormone supplementation (*p* = 0.0169). Meanwhile, both culture samples from estrogen containing medium and samples cultured in estrogen–progesterone containing medium presented with an identical number of clones (*p* = 0.0606 and *p* = 0.3034, respectively). Thus, the highest share of clones (91%) was detected in UL samples cultured with no hormone supplementation. Samples cultured with estrogen or both estrogen and progesterone supplementation demonstrated a lower percentage of clones (70% and 79%, respectively). The least percentage of clones (64%) was registered in UL samples cultured with progesterone supplementation. Therefore, in contrast to samples cultured in the “estrogenic” medium or medium with estrogen–progesterone supplementation, progesterone supplementation in the culture medium is associated with a lower number of karyotypically different clones detected in UL culture samples.

### 2.2. Cytogenetic Characterization of UL Clonal Composition

To shed light on the mechanisms underlying the cytogenetic heterogeneity in ULs, the identified chromosomal aberrations were characterized in terms of their stepwise or one-off acquisition as well as already known models of tumor evolution.

For all eleven ULs showing cytogenetically abnormal clones in the samples cultured with or without hormone supplementation, the ratio of breakpoint junctions to the number of rearranged chromosomes was calculated, and association between the parameters was analyzed. In 8 out of the 11 ULs (UL5–UL12), the average number of breakpoints per rearranged chromosome equaled 2, while 3 other ULs showed higher values of 2.25 in UL15, 2.29 in UL14, and 4.67 in UL13 (Table 1). A positive correlation (ρ = 0.9973, *p* < 0.0001; Spearman coefficient) and strong linear dependency (R^2^ = 0.8386, *p* < 0.0001; linear regression) between the number of breakpoints and the number of rearranged chromosomes was registered. This suggests that the lowest number of breakpoints per rearranged chromosome (=2) and a strong correlation between the number of breakpoints and the number of rearranged chromosomes are common for karyotypically abnormal ULs. In other words, ULs are largely associated with even distributions of breakpoint junctions over rearranged chromosomes, whereas clusterization of breakpoints on a single chromosome is rather uncommon.

Approximately one-third of ULs with an abnormal karyotype had complex chromosomal rearrangements. Follow-up examination of the ULs focused on whether complex chromosomal rearrangements arise in a single event or result from sequential structural rearrangements. Our study checked the relationship between the number of cytogenetic clones identified in UL and such characteristics as the number of breakpoints and the number of rearranged chromosomes, as well as their ratio. We found strong evidence of a correlation and a linear association between the number of clones in ULs, on the one hand, and the number of breakpoints (ρ = 0.784, *p* = 0.0008, Spearman correlation; R^2^ = 0.7045, *p* < 0.0001, linear regression) and the number of rearranged chromosomes (ρ = 0.785, *p* = 0.0008, Spearman correlation; R^2^ = 0.5473, *p* = 0.0016, linear regression), on the other hand. Thus, in a linear pattern, the number of breakpoints and rearranged chromosomes increases with the number of cytogenetic clones in ULs. This suggests that complex chromosomal rearrangements in ULs arise through sequential events rather than one hit.

Next, we investigated the clonal composition of the ULs in terms of evolutionary models already described in ULs: parallel, linear, and branched [17,28,29,30,31,32]. Parallel tumor evolution is characterized by independently emerging new clones, cytogenetically reflected in a completely different chromosomal abnormalities presented in the tumor. Linear evolution implies that new clones arise in the process of follow-up rearrangements in chromosomally abnormal cells. Branched tumor evolution is a combination of the two models. We outlined six ULs (UL9–11, 13–15) containing ≥2 karyotypically abnormal clones to investigate the presumed origin of chromosomal abnormalities. Those included one UL (UL9) containing two clones with deletions of different localizations in the long arm of chromosome 7, which indicates their independent origin; two ULs (UL13, 14) containing clones with partially identical chromosomal rearrangements, which suggests their sequential origin; and two ULs (UL10–11) containing two clones with deletions of different lengths in the long arm of chromosome 7, which could arise either sequentially or independently. UL15 had three abnormal clones with different chromosomal rearrangements—clone 1, showing a complex rearrangement involving chromosomes 7, 10, 13, and both homologues of chromosome 3; clone 2, with terminal deletion of the long arm of chromosome 3 [46,XX,del(3)(q27)]; and clone 3, with an interstitial deletion involving the q12.1–q23.3 region in the long arm of chromosome 16 (Figure 1). In addition, another karyotypically abnormal clone with a combination of the interstitial deletion in the q12.1–q23.3 region of chromosome 16 and the terminal deletion of the q21-qter region in chromosome 3 was found (Figure 1). This suggests that the branched evolutionary model, where chromosomal abnormalities can occur either independently or one by one, resulting in subclones, is most appropriate for the cytogenetic pattern of UL15 (Figure 1). This allows us to conclude that in terms of tumor evolution, the clonal composition in one UL meets criteria for the parallel model, in two ULs it meets the criteria for the linear model, and one in UL it meets the criteria for the branched model.

### 2.3. The Genomic Index for ULs with Chromosomal Abnormalities

We noted that some analyzed ULs had complex clonal compositions and multiple chromosomal rearrangements. Intratumoral clonal diversity and complex chromosomal rearrangements are mostly associated with malignant, rather than benign, tumorigenesis. For uterine neoplasms diagnosed as STUMP (smooth muscle tumors of uncertain malignant potential), a genomic index is used to predict the course of disease and treatment outcomes [33]. The genomic index is calculated based on the array comparative genomic hybridization (aCGH) results using the A^2^/C ratio, where A is the number of genomic alterations and C is the number of altered chromosomes. A genomic index < 10 is associated with a favorable prognosis and low risk of tumor recurrence, while a genomic index ≥ 10 is characteristic of leiomyosarcomas with a poor outcome.

To calculate the genomic index, we selected 11 ULs with chromosomal abnormalities detected through conventional cytogenetic analysis. aCGH was performed on the corresponding uncultured UL samples. In three ULs, no copy number alterations were detected (UL7, 8, 11) (Table 1). In eight ULs, partial monosomies and trisomies were found. Six ULs had deletions in 7q (UL5, 6, 9, 10, 12, 13). UL14 had deletions in the p32.3–p36.33 and q42.13–q44 regions of chromosome 1 and in the p22.2–p22.31 region of chromosome X, as well as duplication in the q23.3–q25.2 region of chromosome 1. UL15 showed deletions in the q13.11–q13.12 and q25.1–q26.1 regions of chromosome 3, the p12.1–p12.3 region of chromosome 7, and the q12.1–q23.3 region of chromosome 16. In general, the obtained aCGH results were concordant with the UL karyotype, except for cases where balanced chromosomal abnormalities and submicroscopic copy number alterations were identified. This is explained by technical differences between the two methods. In addition, in UL11, aCGH failed to detect the deletion in the 7q, which most likely suggests that abnormal clones were scarcely represented in the native tumor sample. Based on the aCGH results, the genomic index was calculated for the eight ULs that harbored copy number alterations. The genomic index equaled “1” in five ULs (UL5, 6, 10, 12, 13), “4” in UL9 and UL14, and “5.3” in UL15.

Overall, aCGH allowed us to detect the copy number alterations in 73% of ULs harboring chromosomal abnormalities revealed through metaphase chromosome analysis. All ULs with copy number alterations showed low genomic index values (<10), which are associated with a favorable disease prognosis.

### 2.4. UL Cytogenomic Findings and Patients’ Medical Data

The UL intertumoral and intratumoral cytogenetic heterogeneity logically provokes considerations regarding the underlying driving factors. Therefore, we attempted to find out whether patients’ medical history and tumor characteristics would correlate with any cytogenomic properties of the investigated ULs, i.e., the genomic index, patterns of chromosomal abnormalities and number of clones, or changes in the clonal composition detected in UL samples cultured in different conditions. Among patients’ medical history parameters, we considered patients’ age, age of menarche, body mass index, length of the menstrual cycle, duration of menstrual bleeding, number of pregnancies, childbirths, abortions, undeveloped pregnancies, smoking, and hormone-containing drug therapy prescribed by an obstetrician/gynecologist throughout their life. Among the tumor characteristics, we took into account solitary/multiple forms of ULs, tumor size and localization, and the timespan from the UL’s detection to myomectomy.

A positive moderate correlation was revealed between patients’ parity status and the UL breakpoints to the rearranged chromosomes ratio (r = 0.518, *p* = 0.048, Spearman correlation). Although other clinical and medical history data did not correlate reliably with any cytogenomic parameters, a few trends were noted. ULs with an abnormal karyotype were more frequently detected in parous women (*p* = 0.1033, Fischer’s exact test). Patients with a history of hormone therapy prescribed by a gynecologist reported a lower number of chromosomal breaks (*p* = 0.0522, Mann–Whitney test) and less frequently presented with a UL with an abnormal karyotype (*p* = 0.1758, Fischer’s exact test). In contrast to multiple ULs, solitary ULs were more prone to chromosomal rearrangements (*p* = 0.0753, Mann–Whitney test), showing a higher probability to detect clones characterized by differential responses to hormone treatment in vitro (*p* = 0.0567, Fischer’s exact test).

Thus, the ratio of breakpoints to rearranged chromosomes in ULs increases with the patient’s parity. To confirm the associations, which did not reach significance, a much larger sample should be analyzed in a separate study.

## 3. Discussion

Our metaphase chromosome analysis of UL cultures revealed a wide spectrum of chromosomal abnormalities, including deletions, translocations, and complex chromosomal rearrangements, which was fully consistent with other existing cytogenetic studies of ULs [13,34,35,36]. Additionally, cells with a 46,XX karyotype were identified in almost every karyotypically abnormal UL analyzed in the present study, thus supporting the widely accepted secondary origin of chromosomal abnormalities as a result of UL karyotype evolution [17,26,37]. Presumably, UL intratumoral cytogenetic heterogeneity is the consequence of parallel, linear, or branched clonal evolution [28,30,32,38]. The investigated ULs with an abnormal karyotype revealed variants of clonal composition, which fit all three evolutionary models. However, it is unclear so far what are the underlying mechanisms that shape the unique cytogenetic UL portrait and whether karyotype differences among the cells of a UL affect their susceptibility to external stimuli and functions.

Our study showed that ULs are predominantly characterized by simple chromosomal rearrangements—deletions or translocations—with two breakpoint junctions involving one or two chromosomes. However, a significant number of karyotypically abnormal ULs (36%) showed complex rearrangements with >2 breakpoints involving two to eight chromosomes. This provokes the following questions about the origins of complex chromosomal rearrangements in ULs. Firstly, where are the breaks most likely to occur—in the same or in different chromosomes? Secondly, how do complex chromosomal rearrangements arise—as a single event or as stepwise structural changes? The study of the UL cytogenetic pattern showed a positive, strong correlation between the number of chromosomal breakpoint junctions and the number of rearranged chromosomes, and the ratio between the two values equals 2 in 73% of cases. These findings suggest that ULs are primarily characterized by two breakpoints in each rearranged chromosome, with an increased number of breakpoint junctions leading to more chromosomes involved in the rearrangements. Both the number of chromosomal breakpoint junctions and the number of rearranged chromosomes augment linearly with the clone number in ULs. Overall, we can conclude that in ULs, complex chromosomal rearrangements are mostly provoked by the accumulation of sequential structural changes in initially intact chromosomes. The chains of structural abnormalities conducive to complex rearrangements involving multiple chromosomes are indicative of chromoplexy, which has never been reported so far for ULs. Chromoplexy refers to chromoanagenesis—complex chromosomal rearrangements associated with multiple DNA breaks provoking complex chromosome reorganization [39,40,41].

It is noteworthy, however, that the number of chromosomal breakpoints to the number of rearranged chromosomes ratio was >2 for a significant number of leiomyomas (27%), reaching 4.67 in UL13. This suggests that in UL cells, complex chromosomal rearrangements may arise from breakpoint clustering within a single chromosome. Extreme cases of such clustering events constitute a different type of chromoanagenesis termed chromothripsis [42]. Unlike chromoplexy, a few publications have already studied the phenomenon of chromothripsis in ULs [14,15,16,17]. Chromothripsis is characterized by multiple rearrangements arising in a single catastrophic event localized in one or a few chromosomes and showing a characteristic pattern of copy number “oscillations” between disomic and monosomic (occasionally, trisomic) states [43]. Thus, in ULs, complex rearrangements arise mostly due to the sequential involvement of intact chromosomes in simple rearrangements, except for some rare cases where complex rearrangements result from multiple simultaneous rearrangements in one or more chromosomes.

Initially, the terms “karyotype evolution”, “intratumoral heterogeneity”, and “chromoanagenesis” were used to characterize the genomes of malignant tumors. Revealing these phenomena in some ULs raises questions about their potential for malignant transformation. The analyzed ULs did not show histological signs of malignancy. In addition to tumor morphology, the genomic index for smooth muscle tumors of uncertain malignant potential (STUMP) was assessed based on aCGH results. Index values < 10 are associated with a good prognosis and low recurrence risk, while values ≥ 10 suggest a poor prognosis [33]. In the obtained ULs, the genomic index did not exceed 8, which corresponds to the data for ULs reported in other publications [33,44]. To compare, the genomic index in leiomyosarcoma ranges from 13.5 to 180 [44]. Thus, no signs of malignancy are exhibited either morphologically or genetically in ULs with complex chromosomal rearrangements or with intratumoral cytogenetic heterogeneity.

Extensive published and ongoing studies have analyzed correlations between genetic alterations in ULs at the gene and chromosome levels, on the one hand, and the myoma’s size, its cellular composition, and its response to various hormone therapies, on the other hand [12,22,23,24,45,46]. Nevertheless, considering the intratumoral heterogeneity of karyotypically abnormal ULs, it is unclear whether cell clones with different karyotypes from the same UL are equally susceptible to external stimuli. A valuable argument in favor of the differential regulation of karyotypically distinct cells in ULs is the fact that the initial proportions of normal and abnormal clones established in native ULs change after in vitro propagation [17]. Some chromosomally abnormal clones undergo positive selection in vitro, while others are subject to negative selection [17]. However, standard UL culture conditions used for cytogenetic studies do not involve supplementation with sex steroids [17,27,29,36,37,47,48]. Currently, studies devoted to the estrogen and progesterone effect on chromosomally abnormal UL cells have been limited to ULs harboring deletions in 7q. When exposed to hormones in vitro, every individual UL tumor demonstrated a similar direction (i.e., an increase or decrease) of changes in the frequency of the clone with the del(7q) in the same direction, with more prominent shifts in cases of exposure to estrogen or progesterone [49].

In addition to the clonal composition of ULs with del(7q), our study evaluated the effect of estrogen and progesterone in vitro on other chromosomal abnormalities. The greatest diversity of cytogenetic clones was found in UL samples cultured under standard conditions without hormone supplementation, as well as under exposure to both estrogen and progesterone. UL culture exposure to progesterone was associated with a significantly decreased number of detected clones. Although statistically insignificant, a decreased number of clones was also observed following estrogen exposure of the cultured UL samples. Our findings suggest that within a single UL, different cell clones with distinct karyotypes differ in their response to sex hormones in vitro. Presumably, the utmost clonal diversity seems to be strongly dependent on the estrogen–progesterone balance, maintained either through the concomitant presence or absence of both hormones in the culture medium. Without additional estrogen, progesterone supplementation in the culture medium leads to reductions in the clonal diversity, hypothetically due to the expansion of the most hormone-sensitive clone/clones. Thus, by maintaining the state of clonal equilibrium, sex steroids can directly impact the UL’s cytogenetic profile. In addition, dramatic changes in estrogen and progesterone levels are capable of provoking chromosomal damage in ULs, which is indirectly evidenced by the striking overrepresentation of chromosomal rearrangements in ULs from parous women shown by Kuisma and colleagues [50] and in the present study.

We should note that our investigation is a pilot study and therefore has certain limitations. Firstly, it included a relatively small sample size (a total of 15 ULs). However, the analysis of individual cells in every UL allowed us to reveal their clonal composition and factors affecting its variation. Although highly labor-intensive, such an investigation is highly informative in contrast to methods based on total DNA extraction from samples. Meticulous cell by cell examination of the karyotype in each sample allowed us to characterize a set of regularities and confirm them statistically. Follow-up studies using a more extensive sample size should confirm our findings and reveal new patterns. Secondly, our study was performed in vitro on 2D cell cultures. Considering the inability of in vitro models to maintain a near-native tumor microenvironment, which may significantly influence the functionality and viability of UL cells, organotypic 3D UL cultures can offer a promising strategy to verify the obtained findings in the future. Thirdly, the metaphase analysis requires only mitotically active cells, thus limiting their number for examination. Therefore, our study elaborates on qualitative clonal composition dynamics for every UL considering the cultivation settings but without focusing on quantitative changes. To overcome this limitation, interphase FISH is required, which uses DNA probes that allow for locating chromosomal abnormalities in non-dividing cells of tumor samples. Fourthly, the changes in clonal dynamics reported for the same tumor and presumably associated with hormone exposure might be driven either by faster proliferation of cells with a particular karyotype or by slower proliferation/nonviability of cells with a different karyotype. Further cell viability assessment, as well as expression analysis of genes involved in proliferation and apoptosis, can shed light on the mechanism underlying the observed changes.

In general, ULs are characterized by both intertumoral and intratumoral cytogenetic heterogeneity, while most chromosomal abnormalities resemble simple structural rearrangements of secondary origin during tumor growth in vivo. In ULs, complex chromosomal rearrangements essentially emerge as a result of incremental chromosomal involvement in the process of clonal evolution. In rare cases, however, complex chromosomal rearrangements may originate from multiple structural changes, which occur simultaneously in a single or a few chromosomes. The ratio of the cells with different karyotypes within a UL is unstable and strongly depends on the estrogen–progesterone balance. The unique spectrum of chromosomal abnormalities harbored by a particular tumor appears to be the most evident explanation behind the unique response of the every karyotypically abnormal UL to hormonal stimuli. The obtained findings allow us to shed light on the evolution of a UL’s cytogenetic profile, allowing us to build a pathway for future studies of individual karyotypically different clones in order to enhance the efficiency of personalized UL treatment strategies.

## 4. Materials and Methods

### 4.1. Materials

A total of 15 histologically confirmed ULs were sampled during myomectomy from 15 female patients who had not received hormone therapy prior to surgery. Patients’ clinical data were available from medical history records. The biospecimens were used for research under written informed consent obtained from all patients. The study was approved by the Ethics Committee of the D.O. Ott Research Institute of Obstetrics, Gynecology and Reproductology (protocol 114, approved 14 December 2021).

### 4.2. UL Cell Cultures

Standard procedures with specific modifications were used to obtain cell cultures from each UL [17,49,51,52]. Briefly, the UL tissue was minced into pieces of 3–5 mm diameter and incubated with collagenase type IV (200 U/mL, Sigma Aldrich, St. Louis, MO, USA) to disaggregate the cells. The dissociated UL cells were planted in T25 Nunc™ EasYFlask™ (ThermoFischer Scientific, Waltham, MA, USA) flasks containing AmnioMAX™-II Complete Medium (ThermoFischer Scientific, USA) and cultured at 37 °C in a humidified 5% CO_2_ atmosphere. The culture medium was replaced with fresh medium two times a week.

For each UL, 4 cell cultures were obtained: (1) without hormone supplementation (standard conditions) or supplemented with (2) estrogen (10^−8^ M); (3) progesterone (10^−6^ M); and (4) estrogen (10^−8^ M) and progesterone (10^−6^ M) together (Sigma Aldrich, USA). Overall, a total of 60 cultures were obtained from 15 ULs. Hormone concentrations in the incubation media corresponded to peak serum estrogen and progesterone levels in non-gravid women during the menstrual cycle. To minimize potential cytotoxic effects, the ethanol solutions of estrogen and progesterone were diluted in the culture medium and added to the cell culture after it reached 40% confluency. At 80% confluency rate, cells were fixed, and cytogenetic preparations were obtained according to the standard procedure [17,27].

### 4.3. Karyotyping of the UL Culture Samples

Karyotyping of the UL culture samples was performed on QFH/AcD-stained metaphase chromosomes using an Axio Imager Z2 microscope (Carl Zeiss Microscopy GmbH, Munich, Germany) and Ikaros V.5.8.9 software (MetaSystems, Altlußheim, Germany). For each UL, 15 to 40 metaphases were analyzed at a resolution of 400–550 bands per haploid chromosome set.

### 4.4. Fluorescence In Situ Hybridization

For chromosome structural analysis, metaphase FISH was performed on preparations from cultured UL cells. Commercial DNA probes were used according to the manufacturer’s recommendations (Abbott Laboratories, Abbott Park, IL, USA; MetaSystems, Heidelberg, Germany). The FISH signals were analyzed using an Axio Imager Z2 microscope (Carl Zeiss Microscopy GmbH) and Isis V.5.8.9 software (MetaSystems).

### 4.5. Array Comparative Genomic Hybridization

For array comparative genomic hybridization (aCGH), the genomic DNA was extracted from UL fragments using the phenol–chloroform method preceded by cell lysis. The extracted and control DNA were both labeled using the Sure Taq DNA Labeling Kit (Agilent, Santa Clara, CA, USA) and hybridized with Agilent G5963A oligonucleotide microarrays according to the manufacturer’s protocols (Agilent, USA).

### 4.6. Statistical Data Processing

GraphPad Prism 6 software (v. 6.01, GraphPad Software, Boston, MA, USA) was used for statistical evaluation at a significance level α = 0.05. Fischer’s exact test was used to compare the number of clones in UL cell cultures. Spearman’s rank correlation coefficient was calculated to assess the strength of relationships across the number of breakpoint junctions, the number of rearranged chromosomes, and the number of clones detected in UL cell cultures. A linear regression model was implemented to justify the linear relationship hypothesis between the variables. Spearman correlation, Fischer’s exact test, and the Mann–Whitney test were used to assess the relationship between ULs’ cytogenetic characteristics and patients’ medical history data.

## Figures and Tables

**Figure 1 ijms-26-00096-f001:**
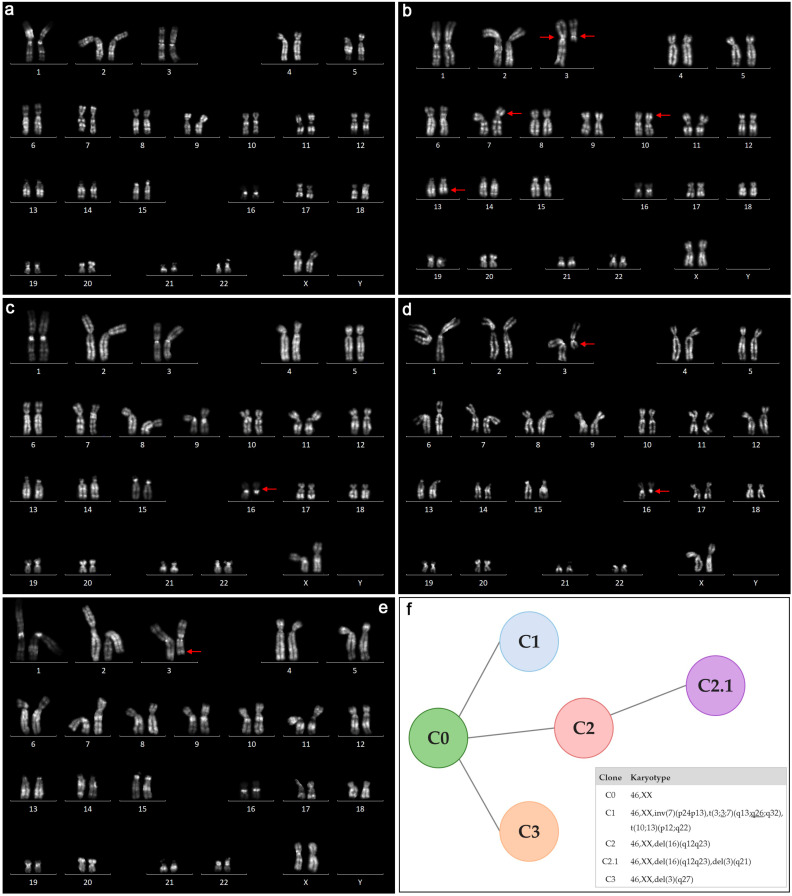
Cytogenetic profile of UL15 revealed through conventional karyotyping. Karyograms (**a**–**e**) demonstrate karyotypes detected in UL cells: 46,XX (**a**); 46,XX,inv(7)(p24p13),t(3;3;7)(q13;q26;q32),t(10;13)(p12;q22) (**b**); 46,XX,del(16)(q12.1q23.3) (**c**); 46,XX,del(16)(q12.1q23.3),del(3)(q21) (**d**); 46,XX,del(3)(q27) (**e**). A schematic explanation of the possible origin of detected cytogenetic clones (**f**). Most probably, three karyotypically abnormal clones (C1, C2, C3) originated independently from 46,XX UL cells (C0) and one clone (C2.1) originated as a subclone of C2 by acquiring additional chromosomal abnormality.

**Table 1 ijms-26-00096-t001:** Cytogenomic characteristics of 15 ULs. Results of conventional cytogenetic analysis are shown for UL samples cultured without hormone supplementation (C), with estrogen (E), with progesterone (P), or with both estrogen and progesterone (E + P) supplementation. The clonal composition of each culture sample is shown using geometric shapes. Green circles correspond to clones with a normal karyotype, while red shapes denote clones with chromosomal abnormalities. Every red geometric shape corresponds to a specific abnormal clone in a given tumor, thus indicating its presence or absence in the culture sample. The order of geometric shapes matches the order of clones in the karyotype designation for a culture sample. For ULs with an abnormal karyotype in at least in one culture sample, array comparative genomic hybridization (aCGH) using genomic DNA from uncultured tumor cells was performed, and the genomic index was calculated.

UL #	Culture Conditions	Metaphase Analysis * of Culture Sample	Schematic Clonal Composition	Number of Detected Clones	Number of Rearranged Chromosomes	Number of Breakpoint Junctions	Ratio of Breakpoint Junctions to the Number of Rearranged Chromosomes	aCGH	Genomic Index
1	C	46,XX[15]		1	0	0	not applicable	not performed	not applicable
E	46,XX[15]	
P	46,XX[15]	
E + P	46,XX[18]	
2	C	46,XX[12]		1	0	0	not applicable	not performed	not applicable
E	46,XX[15]	
P	46,XX[15]	
E + P	46,XX[15]	
3	C	46,XX[18]		1	0	0	not applicable	not performed	not applicable
E	46,XX[33]	
P	46,XX[17]	
E + P	46,XX[13]	
4	C	46,XX[14]		1	0	0	not applicable	not performed	not applicable
E	46,XX[18]	
P	46,XX[15]	
E + P	46,XX[19]	
5	C	46,XX,del(7)(q21q31),t(2;5)(p13;p13)[24]/46,XX[7]	 	2	3	6	2	arr[GRCh37] 7q21.3q31.2(95049263_115448961)×1[0,6]	1
E	46,XX,del(7)(q21q31),t(2;5)(p13;p13)[20]/46,XX[9]	 
P	46,XX,del(7)(q21q31),t(2;5)(p13;p13)[23]/46,XX[8]	 
E + P	46,XX,del(7)(q21q31),t(2;5)(p13;p13)[20]/46,XX[3]	 
6	C	46,XX,t(4;12)(p14;q21)[2]/46,XX[13]	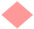 	2	2	4	2	arr[GRCh37] 7q21.3q32.1(94547135_127595933)×1[0,6]	1
E	46,XX,t(4;12)(p14;q21)[24]/46,XX[30]	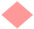 
P	46,XX,t(4;12)(p14;q21)[16]/46,XX[12]	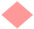 
E + P	46,XX,t(4;12)(p14;q21)[14]/46,XX[11]	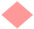 
7	C	46,XX,t(1;7)(p36;q22)[5]/46,XX[25]	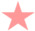 	2	2	4	2	arr(X,1-22)×2	not applicable
E	46,XX,t(1;7)(p36;q22)[1]/46,XX[7]	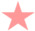 
P	46,XX[14]	
E + P	46,XX,t(1;7)(p36;q22)[3]/46,XX[15]	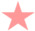 
8	C	46,XX,t(6;17)(p21;p13)[2]/46,XX[13]	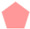 	2	2	4	2	arr(X,1-22)×2	not applicable
E	46,XX[18]	
P	46,XX[30]	
E + P	46,XX[24]	
9	C	46,XX,del(7)(q11.23q31.1)[4]/46,XX,del(7)(q21.1q36.1)[2]/46,XX[34]	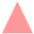 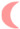 	3	2	4	2	arr[GRCh37] 7q11.23q31.1(76200157_111991020)×1[0,7],7q31.1q36.1(112933041_149983247)×1[0,4]	4
E	46,XX,del(7)(q21.1q36.1)[14]/46,XX,del(7)(q11.23q31.1)[11]	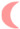 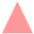
P	46,XX,del(7)(q11.23q31.1)[42]	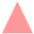
E + P	46,XX,del(7)(q21.1q36.1)[23]/46,XX,del(7)(q11.23q31.1)[1]	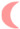 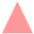
10	C	46,XX,del(7)(q21q36)[2]/46,XX[15]	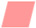 	3	2	4	2	arr[GRCh37] 7q22.1q31.1(98667421_113540247)×1[0,7]	1
E	46,XX[13]	
P	46,XX,del(7)(q21q31)[1]/46,XX[15]	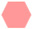 
E + P	46,XX,del(7)(q21q31)[6]/46,XX[7]	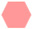 
11	C	46,XX,del(7)(q21q31)[12]/46,XX,del(7)(q21q32)[7]/46,XX[7]	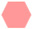 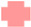 	3	2	4	2	arr(X,1-22)×2	not applicable
E	46,XX,del(7)(q21q31)[6]/46,XX[8]	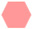 
P	46,XX,del(7)(q21q31)[9]/46,XX[7]	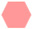 
E + P	46,XX,del(7)(q21q31)[2]/46,XX,del(7)(q21q32)[1]/46,XX[8]	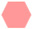 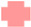 
12	C	46,XX,del(7)(q21q32)[11]/46,XX[1]	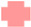 	2	1	2	2	arr[GRCh37] 7q21.13q32.2(89082865_129330045)×1[0,9]	1
E	46,XX,del(7)(q21q32)[15]	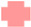
P	46,XX,del(7)(q21q32)[16]	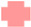
E + P	46,XX,del(7)(q21q32)[6]/46,XX[1]	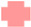 
13	C	46,XX,del(7)(q21q31),der(12)(pter→p11.2::q22→q23::p11.2→q14::q23→q24::q21→q14::q24→qter)[22]/46,idem,dup(16)(q24q11)[2]	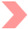 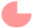	2	3	14	4.67	arr[GRCh37] 7q21.3q31.1(95640736_112487227)×1[0,8]	1
E	46,XX,del(7)(q21q31),der(12)(pter→p11.2::q22→q23::p11.2→q14::q23→q24::q21→q14::q24→qter)[21]	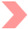
P	46,XX,del(7)(q21q31),der(12)(pter→p11.2::q22→q23::p11.2→q14::q23→q24::q21→q14::q24→qter)[16]	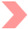
E + P	46,XX,del(7)(q21q31),der(12)(pter→p11.2::q22→q23::p11.2→q14::q23→q24::q21→q14::q24→qter)[10]	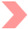
14	C	45,XX,der(1)(1p32→1q12::21q11.2→21qter),der(2)(2pter→2q31::3p24→3pter),der(3)(2qter→2q31::3p24→3qter),der(3)(3pter→3q25::19q13→19qter),der(6)(3qter→3q25::6p21→6qter),der(8)(8pter→8q10::1q42→1q12),der(19)(19pter→19q13::6p21→6pter),-21[1]/45,XX,der(1)(8qter→8q12::1p32→1q12::21q11.2→21qter),der(2)(2pter→2q31::3p24→3pter),der(3)(2qter→2q31::3p24→3qter),der(3)(3pter→3q25::19q13→19qter),der(6)(3qter→3q25::6p21→6qter),der(8)(8pter→8q10::1q42→1q12),der(19)(19pter→19q13::6p21→6pter),-21[9]/46,XX[4]	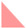 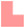 	3	7	16	2.29	arr[GRCh37] 1p36.33p32.3(746608_54801890)×1[0,9],1q42.13q44(228309494_248836709)×1[0,9]	4
E	45,XX,der(1)(8qter→8q12::1p32→1q12::21q11.2→21qter),der(2)(2pter→2q31::3p24→3pter),der(3)(2qter→2q31::3p24→3qter),der(3)(3pter→3q25::19q13→19qter),der(6)(3qter→3q25::6p21→6qter),der(8)(8pter→8q10::1q42→1q12),der(19)(19pter→19q13::6p21→6pter),-21[16]	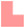
P	45,XX,der(1)(8qter→8q12::1p32→1q12::21q11.2→21qter),der(2)(2pter→2q31::3p24→3pter),der(3)(2qter→2q31::3p24→3qter),der(3)(3pter→3q25::19q13→19qter),der(6)(3qter→3q25::6p21→6qter),der(8)(8pter→8q10::1q42→1q12),der(19)(19pter→19q13::6p21→6pter),-21[13]	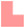
E + P	45,XX,der(1)(1p32→1q12::21q11.2→21qter),der(2)(2pter→2q31::3p24→3pter),der(3)(2qter→2q31::3p24→3qter),der(3)(3pter→3q25::19q13→19qter),der(6)(3qter→3q25::6p21→6qter),der(8)(8pter→8q10::1q42→1q12),der(19)(19pter→19q13::6p21→6pter),-21[1]/45,XX,der(1)(8qter→8q12::1p32→1q12::21q11.2→21qter),der(2)(2pter→2q31::3p24→3pter),der(3)(2qter→2q31::3p24→3qter),der(3)(3pter→3q25::19q13→19qter),der(6)(3qter→3q25::6p21→6qter),der(8)(8pter→8q10::1q42→1q12),der(19)(19pter→19q13::6p21→6pter),-21[10]	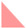 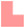
15	C	46,XX,der(3)(3pter→3q13::7q32→7qter),der(3)(3pter→3q26::3q13→3qter),der(7)(7pter→7p24::7p13→7p24::7p13→7q32::3q26→3qter),der(10)(13qter→13q22::10p12→10qter),der(13)(13pter→13q22::10p12→10pter)[11]/46,XX,del(16)(q12.1q23.3)[5]/46,XX[4]	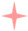 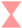 	5	8	18	2.25	arr[GRCh37] 3q13.11q13.12(103804500_107433564)×1[0.3],3q25.1q26.1(151099000_168317000)×1[0.3],7p12.3p12.1(46770947_51487239)×1[0.3],16q12.1q23.3(49027670_84191357)×1[0,5]	5.3
E	46,XX,der(3)(3pter→3q13::7q32→7qter),der(3)(3pter→3q26::3q13→3qter),der(7)(7pter→7p24::7p13→7p24::7p13→7q32::3q26→3qter),der(10)(13qter→13q22::10p12→10qter),der(13)(13pter→13q22::10p12→10pter)[8]/46,XX,del(16)(q12.1q23.3)[6]/46,XX,del(16)(q12.1q23.3),del(3)(q21)[2]/46,XX[2]	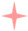 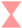 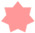 
P	46,XX,del(16)(q12.1q23.3)[19]/46,XX,del(3)(q27)[13]/46,XX[11]	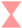 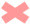 
E + P	46,XX,der(3)(3pter→3q13::7q32→7qter),der(3)(3pter→3q26::3q13→3qter),der(7)(7pter→7p24::7p13→7p24::7p13→7q32::3q26→3qter),der(10)(13qter→13q22::10p12→10qter),der(13)(13pter→13q22::10p12→10pter)[9]/46,XX,del(16)(q12.1q23.3)[3]/46,XX[10]	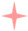 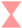 

* Most karyotypes are described with the short system of karyotype designation. Cases of complex rearrangements are described, where appropriate, with the detailed system.

## Data Availability

The original contributions presented in this study are included in the article. Further inquiries can be directed to the corresponding author.

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
