# Peer review of "In Vitro Effect of Estrogen and Progesterone on Cytogenetic Profile of Uterine Leiomyomas"

_ijms, 2024, doi:10.3390/ijms26010096_

Round 1

Reviewer 1 Report

Comments and Suggestions for Authors

The article you provided explores unique findings in uterine leiomyomas (ULs) and attempts to fill several field-specific gaps. The following aspects appear both original and significant in addressing these gaps. The study advances understanding by analyzing the variation in chromosomal abnormalities within individual ULs. The researchers used various culturing conditions (without hormones, with estrogen, with progesterone, and with both) to assess how hormone presence influences karyotype diversity. This approach highlights the complex clonal evolution of ULs, especially under hormonal influences, and represents a novel perspective on how chromosomal abnormalities might impact the tumor's response to treatments. The authors of the study should consider several methodological improvements to enhance the robustness of their findings:

1.Expanding the sample size from 15 to a larger cohort could improve the statistical power and generalizability of the findings, allowing for more conclusive results across varied cases of uterine leiomyomas (ULs).

2.Currently, the study examines four culture conditions standard, estrogen, progesterone, and combined estrogen-progesterone. Additional hormone concentration levels or varying the timing of hormone exposure could reveal more nuanced responses of UL cell clones to hormonal changes.

3.While the study focuses on chromosomal abnormalities under hormone exposure, analyzing the effects of estrogen and progesterone on cell viability and proliferation rates could provide a more comprehensive understanding of hormonal impact on UL cells beyond cytogenetic effects.

These methodological improvements could strengthen the study’s findings and help determine more precise therapeutic targets for personalized treatments.

Author Response

Comment 1. [Expanding the sample size from 15 to a larger cohort could improve the statistical power and generalizability of the findings, allowing for more conclusive results across varied cases of uterine leiomyomas (ULs).]

Response 1. [We would like to thank the Reviewer for pointing this out. We agree that a larger sample size would allow us to make additional cytogenetic findings and identify more patterns regarding the effect of sex steroid hormones on uterine leiomyomas (ULs) with chromosomal abnormalities, as well as their karyotype evolution. The relatively small number of ULs enrolled in the study is primarily due to the complexity of the methodological approach, namely, cell-by-cell analysis. Cell-by-cell analysis, which was applied to characterize the cytogenetic portrait of individual ULs, is highly informative and allowed us to identify novel and intriguing findings even on a small number of ULs. Including of new samples to the study would take months of research. Instead, we have added a limitation paragraph to the Discussion (lines 376-383).]

Comment 2. [Currently, the study examines four culture conditions standard, estrogen, progesterone, and combined estrogen-progesterone. Additional hormone concentration levels or varying the timing of hormone exposure could reveal more nuanced responses of UL cell clones to hormonal changes.]

Response 2. [Thank you for this comment.  The experiments with varying concentrations of estrogen and progesterone could shed light on the mechanisms of hormonal regulation of UL growth and simulate a woman's hormonal status at different periods of life. However, our study was designed for a pilot testing of the sex steroid hormone effect on cytogenetically heterogeneous cells from a single UL, since this had not previously been studied for ULs with different chromosomal abnormalities. Varying hormone concentration levels and/or timing of hormone exposure are the subjects of future studies.]

Comment 3. [While the study focuses on chromosomal abnormalities under hormone exposure, analyzing the effects of estrogen and progesterone on cell viability and proliferation rates could provide a more comprehensive understanding of hormonal impact on UL cells beyond cytogenetic effects.]

Response 3. [Thank you for this comment.  The additional methods for assessing cell viability, as well as the expression levels of genes responsible for proliferation and apoptosis, will clarify the link between UL growth, sex steroid hormones and chromosomal abnormalities. We will keep it in mind when designing next research.

We indicated in the Discussion section that the inability to determine the exact cause of the observed shifts in the clonal composition in response to hormone exposure is a limiting factor in our study (kindly see lines 392-397).

We are very grateful to the Reviewer for valuable comments on our work and suggestions for future research.]

Reviewer 2 Report

Comments and Suggestions for Authors

A manuscript entitled “In Vitro Effect of Estrogen and Progesterone on Cytogenetic Profile of Uterine Leiomyomas” addresses the important issue of investigating the intratumoral karyotype diversity of uterine leiomyomas (ULs).

I found the introduction well-written, with sufficient details to introduce the reader to the topic. The methods are well described and supported by well-explained results, followed by figures and tables.

Figure 1 – in the caption please explain the difference between the red shapes used in the figure

This complex matter is easy to follow throughout the results and discussion section. The obtained results are important, even though the number of samples is relatively small, but as the authors stated themselves further studies with higher sample sizes would be of more significance. The statistical tests used to detect the significance of the obtained results are well chosen.

The references are well chosen, even though over ten are 30 years old. I am sure much has been done in this area since then, so maybe the newest references can be cited. Also, more than 1/5 of the papers cited are from the authors themselves. On the other hand, the topic is in the field of the authors' expertise, it is not unusual to refer to your previous work. 

A few corrections in the text are proposed:

Line 63 – FISH; when mentioned the first time in the text state the full name, a short should be used after

Lines 243-245 use the same pattern (letters or numbers) when explaining the genomic index

“The genomic index equaled one in 5 ULs (UL5, 6, 10, 12, 13), four in UL9 and UL14, and 5.3 in UL15.“

Author Response

Comment 1. [Figure 1 – in the caption please explain the difference between the red shapes used in the figure]

Response 1. [Authors would like to thank the Reviewer for pointing this out. We corrected the caption of the Figure 1 (kindly see lines 145-149).]

Comment 2. [This complex matter is easy to follow throughout the results and discussion section. The obtained results are important, even though the number of samples is relatively small, but as the authors stated themselves further studies with higher sample sizes would be of more significance. The statistical tests used to detect the significance of the obtained results are well chosen.]

Response 2. [We agree with the comment. We additionally indicated in the Discussion section that a small sample size is a limiting factor in our study (kindly see lines 376-383).]

Comment 3. [The references are well chosen, even though over ten are 30 years old. I am sure much has been done in this area since then, so maybe the newest references can be cited. Also, more than 1/5 of the papers cited are from the authors themselves. On the other hand, the topic is in the field of the authors' expertise, it is not unusual to refer to your previous work.]

Response 3. [Thank you for the comment. Some old references were included as they represent fundamental studies in UL cytogenetics. State-of-the-art studies were also cited. Self-citing was included where necessary, because we have a series of UL studies and already have obtained important results preceding or explaining findings of the present study.]

Comment 4. [Line 63 – FISH; when mentioned the first time in the text state the full name, a short should be used after]

Response 4. [We would like to thank the Reviewer for pointing this out. We have now made a correction (kindly see line 63).]

Comment 5. [Lines 243-245 use the same pattern (letters or numbers) when explaining the genomic index

 “The genomic index equaled one in 5 ULs (UL5, 6, 10, 12, 13), four in UL9 and UL14, and 5.3 in UL15.“]

Response 5. [We agree with the comment. We have now made an appropriate correction (kindly see lines 249-250).

We are very grateful to the Reviewer for valuable comments on our work and suggestions for future research.]

Round 2

Reviewer 2 Report

Comments and Suggestions for Authors

Dear Authors,

   Thank you for accepting my comments and recommendation for improving your paper.

As for my comment to explain the difference between the red shapes used in the figure you added the following sentence:

Line 145-149:  “Green circles correspond to clones with 46,XX karyotype, while red shapes (square, rhombus, pentagon or star) denote clones with chromosomal abnormalities. The number of red shapes per every UL equals the number of detected abnormal clones. Every geometric shape corresponds to a specific abnormal clone in a given tumor, thus indicating its presence or absence in the culture sample.”

 which explains the difference between used shapes, but as explained the meaning of the green shape (46,XX karyotype) please give the meaning of each red shape. Especially since you added the following sentence:

Line 377-378:  However, the analysis of individual cells in every UL allowed us to reveal their clonal composition and factors affecting their variation.

Due to the length of the individual karyotype (UL 13-15), you can maybe add a column in Table 1 with certain green and red shapes.

Otherwise, I think this paper is well worth publishing. 

Author Response

Comment 1. [As for my comment to explain the difference between the red shapes used in the figure you added the following sentence:

Line 145-149:  “Green circles correspond to clones with 46,XX karyotype, while red shapes (square, rhombus, pentagon or star) denote clones with chromosomal abnormalities. The number of red shapes per every UL equals the number of detected abnormal clones. Every geometric shape corresponds to a specific abnormal clone in a given tumor, thus indicating its presence or absence in the culture sample.”

 which explains the difference between used shapes, but as explained the meaning of the green shape (46,XX karyotype) please give the meaning of each red shape. Especially since you added the following sentence:

Line 377-378:  However, the analysis of individual cells in every UL allowed us to reveal their clonal composition and factors affecting their variation.

Due to the length of the individual karyotype (UL 13-15), you can maybe add a column in Table 1 with certain green and red shapes.]

Response 1. [Thank you for the comment and for suggestion to present geometric shapes in Table 1. We have now included all shapes illustrating detected clones in the table – kindly see ‘Schematic clonal composition’. Moreover, we now use different shapes for every abnormal clone. Order of shapes corresponds to the order of clones in the karyotype designation. We removed the Figure 1 from the manuscript. Corrections are marked with green.

We are very grateful to the Reviewer for helpful suggestions improving our work.]

Round 3

Reviewer 2 Report

Comments and Suggestions for Authors

Dear Authors,

   Thank you for accepting my suggestions.